# Prediction of Acute Respiratory Distress Syndrome in Traumatic Brain Injury Patients Based on Machine Learning Algorithms

**DOI:** 10.3390/medicina59010171

**Published:** 2023-01-15

**Authors:** Ruoran Wang, Linrui Cai, Jing Zhang, Min He, Jianguo Xu

**Affiliations:** 1Department of Neurosurgery, West China Hospital, Sichuan University, Chengdu 610041, China; 2Institute of Drug Clinical Trial·GCP, West China Second University Hospital, Sichuan University, Chengdu 610041, China; 3Diseases of Women and Children, Sichuan University, Ministry of Education, Chengdu 610041, China; 4Department of Critical Care Medicine, West China Hospital, Sichuan University, Chengdu 610041, China

**Keywords:** traumatic brain injury, acute respiratory distress syndrome, machine learning, prognosis factors

## Abstract

***Background*:** Acute respiratory distress syndrome (ARDS) commonly develops in traumatic brain injury (TBI) patients and is a risk factor for poor prognosis. We designed this study to evaluate the performance of several machine learning algorithms for predicting ARDS in TBI patients. ***Methods*:** TBI patients from the Medical Information Mart for Intensive Care-III (MIMIC-III) database were eligible for this study. ARDS was identified according to the Berlin definition. Included TBI patients were divided into the training cohort and the validation cohort with a ratio of 7:3. Several machine learning algorithms were utilized to develop predictive models with five-fold cross validation for ARDS including extreme gradient boosting, light gradient boosting machine, Random Forest, adaptive boosting, complement naïve Bayes, and support vector machine. The performance of machine learning algorithms were evaluated by the area under the receiver operating characteristic curve (AUC), sensitivity, specificity, accuracy and F score. ***Results*:** 649 TBI patients from the MIMIC-III database were included with an ARDS incidence of 49.5%. The random forest performed the best in predicting ARDS in the training cohort with an AUC of 1.000. The XGBoost and AdaBoost ranked the second and the third with an AUC of 0.989 and 0.815 in the training cohort. The random forest still performed the best in predicting ARDS in the validation cohort with an AUC of 0.652. AdaBoost and XGBoost ranked the second and the third with an AUC of 0.631 and 0.620 in the validation cohort. Several mutual top features in the random forest and AdaBoost were discovered including age, initial systolic blood pressure and heart rate, Abbreviated Injury Score chest, white blood cells, platelets, and international normalized ratio. ***Conclusions*:** The random forest and AdaBoost based models have stable and good performance for predicting ARDS in TBI patients. These models could help clinicians to evaluate the risk of ARDS in early stages after TBI and consequently adjust treatment decisions.

## 1. Introduction

Traumatic brain injury (TBI) is a widely concerning health issue causing a huge burden to society. It has been estimated that 69 million people suffer TBI annually around the world [1]. The poor prognosis of TBI is not only attributable to the severity of intracranial injury and concomitant trauma of the extracranial region but is also caused by various complicated organ dysfunctions such as acute kidney injury, coagulopathy, respiratory failure and acute respiratory distress syndrome (ARDS) [2]. Previous studies have shown that ARDS was a common pulmonary complication in TBI patients with the incidence ranging from 1% to 60% [2,3]. ARDS also has been confirmed as a risk factor for poor prognosis including higher mortality, poorer neurological outcome and longer length of hospital stay in some studies [4,5,6,7]. Studies have explored risk factors for ARDS in TBI patients including younger age, male sex, admission tachycardia, underlying respiratory and vascular diseases, pneumonia, head AIS, early crystalloids, early platelet transfusion and intracranial hypertension [6,7,8]. While one recent meta-analysis indicated age, male gender, white race, head AIS, Marshall CT score, GCS on admission, and increased intracranial pressure during hospitalizations were not significant predictors for ARDS in TBI [9]. Exploring potential risk factors for ARDS after TBI and identifying patients with a higher risk for ARDS in the early stage after injury is important for clinicians to devise optimal treatment strategies including setting appropriate parameters on the ventilator. Trying to avoid the development or progression of ARDS in clinical practice may improve the prognosis of TBI patients. There is no study developing a model to evaluate the risk of ARDS in TBI patients. Machine learning algorithms perform well on predicting outcome events for patients based on their advantages in dealing with complex data and nonlinear relationships. We designed this study to evaluate the performance of different machine learning algorithms when predicting ARDS in TBI patients.

## 2. Methods and Materials

### 2.1. Patients

This study included patients derived from the Medical Information Mart for Intensive Care-III (MIMIC-III) database. Produced by the computational physiology laboratory of Massachusetts Institute of Technology (MIT) (Cambridge, MA, USA), this freely available database collects electronic medical records of patients hospitalized in the Beth Israel Deaconess Medical Center (BIDMC) (Boston, MA, USA) between 2001 and 2012 and receives ethical approval from the institutional review boards of MIT and BIDMC. Patients in the MIMIC-III were deidentified and anonymized to protect personal privacy. Our study extracted patients diagnosed with TBI from the MIMIC-III based on ICD-9 codes (80000-80199; 80300-80499; 8500-85419). Some of the TBI patients were excluded from this study if they met the following criteria: (1) Age < 18; (2) Lacked records of Glasgow Coma Scale (GCS) on admission; (3) Lacked records of vital signs and laboratory tests; (4) Abbreviated Injury Score (AIS) head < 3; (5) Lacked records of arterial oxygen pressure (PaO_2_) and corresponding fraction of inspired oxygen (FiO_2_) (Figure 1). A total of 649 TBI patients were finally included in our study. The study was designed and conducted to comply with the ethical standards of the Helsinki declaration. The study design was approved by the ethical committee of West China hospital (2021-1598).

### 2.2. Study Variables

Age, gender and underlying diseases including diabetes, hypertension, hyperlipidemia, coronary heart disease, liver disease, chronic renal disease, and malignancy were included. Initial vital signs including systolic blood pressure (SBP), diastolic blood pressure (DBP), heart rate, and respiratory rate were recorded. Glasgow Coma Score (GCS), Abbreviated Injury Score (AIS) of chest, and Injury Severity Score (ISS) were collected to reflect the severity of injuries. Laboratory tests analyzed from the first blood sample since admission were selected as features including white blood cells (WBCs), platelets, red blood cells (RBCs), hemoglobin, glucose, blood urea nitrogen, serum creatinine, serum sodium, serum potassium, serum chloride, serum calcium, prothrombin time, and international normalized ratio (INR). Initial ventilation related parameters including PaO_2_, FiO_2_, PaO_2_/FiO_2_ ratio were extracted. Intracranial injury locations were fetched including epidural hematoma (EDH), subdural hematoma (SDH), subarachnoid hemorrhage (SAH), and intraparenchymal hemorrhage (IPH). Medical treatments during the first day since admission were collected, including RBC transfusion, platelet transfusion, anticoagulant use, antiplatelet use and vasopressor use. Records of mechanical ventilation and neurosurgical operation were collected. A total of 40 features were finally included in the process of developing machine learning models. The outcome of this study was the development of ARDS which was diagnosed based on the Berlin definition [10].

### 2.3. Statistical Analysis

The normality of collected variables was determined by the Kolmogorov–Smirnov test. Variables with normal distribution and non-normal distribution were presented as mean ± standard deviation and median (interquartile range), respectively. Categorical variables were shown as counts (percentage). Differences of collected continuous variables between ARDS group and non-ARDS group were analyzed by the Student’s *t*-test and Mann–Whitney U test. Differences in collected categorical variables between two groups were analyzed by the Chi-square test or the Fisher exact test. *p* < 0.05 was considered as being statistically significant.

### 2.4. Machine Learning Algorithms

TBI patients included from the MIMIC-III dataset were randomly divided into the training set and the validation set with a ratio of 7:3. There were six machine learning algorithms that were trained with five-fold cross validation in the training set to predict ARDS including extreme gradient boosting (XGBoost), light gradient boosting machine (light GBM), random forest, adaptive boosting (AdaBoost), complement naïve Bayes, and support vector machine (SVM). The optimal parameters of each machine learning algorithm were automatically explored during the cross-validation process. Trained machine learning predictive models were then verified in the validation set by evaluating multiple indexes including area under the receiver operating characteristics curve (AUC), accuracy, sensitivity, specificity, positive predicted value (PPV), negative predicted value (NPV) and F1 score. The Shapley Additive explanation (SHAP) method was utilized for evaluation of the feature importance in machine learning predictive models and visualized explanation of predictive models. All statistical analyses and figures were performed using the Extreme smart analysis—an online statistical analysis platform based on the Python (Amsterdam, The Netherlands).

## 3. Results

### 3.1. Comparison between Final Included Patients and Those Lacking Records of PaO_2_ and FiO_2_

A total of 2031 patients were excluded from this study due to the following criteria: (1) age < 18 (*n* = 32); (2) lacked records of GCS on admission (*n* = 65); (3) lacked records of vital signs and laboratory tests (*n* = 116); (4) AIS head < 3 (*n* = 187); (5) lacked records of PaO_2_ and corresponding FiO_2_ (*n* = 1631) (Figure 1). Finally, 649 TBI patients were included with the ARDS incidence being 49.5%. A large number of TBI patients (1631/2680, 60.86%) were excluded due to a lack of records of PaO_2_ and corresponding FiO_2_. The comparison between these patients and final included patients were shown in Appendix A. Compared with these excluded patients, final included patients were mainly severe TBI (GCS: 6 (3–9), median (quartiles)) and had younger age (59.3 vs. 66.8, *p* < 0.001). The incidence of RBC transfusion (*p* < 0.001), antiplatelet transfusion (*p* < 0.001), vasopressor use (*p* < 0.001), mechanical ventilation (*p* < 0.001) and neurosurgery (*p* < 0.001) were higher in the final included patients. They also had higher mortality than those excluded patients (29.4% vs. 13.1%, *p* < 0.001). These differences indicated the included population of this study was mainly severe TBI.

### 3.2. Baseline Characteristics of Included TBI Patients

Among included patients, the age of the ARDS group was higher than the non-ARDS group (*p* = 0.027) (Table 1). Comorbidities did not significantly differ between the ARDS group and the non-ARDS group. The ARDS group had higher AIS chest (*p* < 0.001) and ISS (*p* = 0.009) than the non-ARDS group. The GCS did not show significant difference between the two groups (*p* = 0.724). Laboratory tests showed that platelets (*p* = 0.004) were lower in the ARDS group while prothrombin time (*p* = 0.002) and INR (<0.001) were higher in the ARDS group. The initial PaO_2_ (*p* < 0.001) was lower in the ARDS group while the initial FiO_2_ did not show significant difference between two groups (*p* = 0.874). The initial PaO_2_/FiO_2_ ratio of the non-ARDS group and the ARDS group were 356 and 248, respectively. The percentage of mild, moderate and severe ARDS among overall ARDS patients was 43.5%, 39.7% and 16.7%, respectively (Figure 2). Compared with the non-ARDS group, the ARDS group was more likely to receive platelet transfusion during the first day (*p* = 0.0027). Finally, the ARDS group had a longer length of ICU stay (<0.001) and length of hospital stay (<0.001).

### 3.3. Performance of Machine Learning Algorithms for Predicting ARDS in TBI

The random forest performed the best on predicting ARDS in the training cohort with an AUC value of 1.000 (Table 2) (Figure 3A). The accuracy, sensitivity, specificity, PPV, NPV, F1 score of the random forest in the training cohort was 0.998, 1.000, 1.000, 1.000, 0.997 and 1.000, respectively. The XGBoost and AdaBoost ranked second and third with an AUC of 0.989 and 0.815. The random forest still performed the best in predicting ARDS in the validation cohort with an AUC value of 0.652 (Table 3) (Figure 3B). The accuracy, sensitivity, specificity, PPV, NPV, and F1 score of the random forest in the validation cohort was 0.542, 0.719, 0.579, 0.767, 0.526, 0.716, respectively. The AdaBoost and XGBoost ranked second and third with an AUC of 0.631 and 0.620. Generally, the random forest performed well and stably in predicting ARDS both in the training cohort and the validation cohort. The AdaBoost is second only to the random forest while the XGBoost showed significantly different performance between the training cohort and the validation cohort.

### 3.4. Important Features in Machine Learning Algorithms for Predicting ARDS in TBI

Feature importance derived from random forest and AdaBoost are shown in Figure 4A and Figure 4B, respectively. The SHAP value of all patients’ output in the random forest model and the AdaBoost model are presented in Figure 4C,D. The common important features in the random forest and the AdaBoost were analyzed. Figure 5A showed 15 common features were discovered among the top 20 features in these two algorithms including platelet, INR, AIS chest, heart rate, DBP, WBC, age, serum chloride, hemoglobin, SBP, SDH, respiratory rate, serum sodium, GCS, and RBC. Figure 5B showed seven common features were discovered among the top ten features in these two algorithms including platelet, INR, AIS chest, heart rate, WBC, age, SBP.

## 4. Discussion

The incidence of ARDS in the study was 49.5%, which was similar to the previously reported incidence of ARDS in TBI ranging from 1% to 60% [2,3,9]. The actual incidence of ARDS in TBI patients from the MIMIC-III database may be lower than 49.5% because a large number of TBI patients that lacked relevant records were excluded from this study. The significant variation in the reported incidence of ARDS in TBI may be attributable to the difference of injury severity, treatment strategy and healthcare level in different medical centers. Compared with the non-ARDS group, the ARDS group in the study had a longer length of ICU stay and length of hospital stay. The 30-day mortality did not show statistical significance between these two groups. This fact is contradictory to the finding of one meta-analysis which indicated the survival proportion was significantly higher in TBI patients without ARDS than those with ARDS [9]. The insignificance of survival between these two groups in our study may be caused by the exclusion of a large number of mild TBI patients. Due to the high prevalence of ARDS and poor its prognosis, exploring risk factors for ARDS and evaluating the risk of developing ARDS in the early phase after TBI is necessary to decrease the possibility of developing ARDS and to improve the prognosis of TBI patients.

In this study, the random forest and AdaBoost achieved good and stable performances in predicting ARDS, both in the training cohort and the validation cohort among several machine learning algorithms. Trained based on the bagging method, the random forest is an ensemble classifier composed of multiple decision trees. It integrates all classified voting results of individual trees and judges the category with the most votes as the final output. The boosting method means combining many weak classifiers to produce a powerful classifier to improve the predictive accuracy of the final model. As a classical boosting algorithm, Adaboost has a high detection rate and is not prone to over fitting. A total of seven mutual features were discovered among the top ten features in random forest and adaboost including platelet, INR, AIS chest, heart rate, WBC, age, and SBP. The platelet and INR are essential components of the coagulation test. Previous studies showed coagulative disorders are prevalent in TBI with the incidence of coagulopathy ranging from 13% to 54% [11,12,13,14,15]. Actually, the coagulation system plays an important role in the pathophysiological process of ARDS [16,17]. The imbalance between inflammation and coagulation leads to an inflammatory response, formation of microthrombi and diffused deposition of fibrin in pulmonary capillary bed and alveoli [18,19]. As a key element in ARDS development, the process of immune-thrombosis formation involves many kinds of cells including platelets, neutrophils, endothelial cells [18]. Correspondingly, the WBC is another important feature for predicting ARDS in our developed random forest and adaboost models.

In addition to the coagulation indexes and WBC, heart rate and SBP which may collectively reflect the tissue perfusion were also important features in machine learning based models. The shock status would undoubtedly decrease the transport of blood and oxygen to pulmonary tissue and accelerate lung injury. Finally, AIS chest and age were important features in machine learning based models. One previous study confirmed rib fracture as a risk factor for ARDS after mild TBI [20]. The thoracic trauma may cause direct mechanical damage to the pulmonary tissue or increase the risk of pneumonia by restricting respiratory amplitude. One epidemiological research study with a large sample size found younger age was significantly associated with the higher risk of ARDS in isolated severe TBI patients, while another study showed that elderly trauma patients had a higher risk of ARDS than non-elderly trauma patients [7,21]. The influence of age on ARDS occurrence and the corresponding mechanism in TBI patients is still worth investigating. Composed of these above-mentioned important features, random forest or adaboost based models may be effective in predicting the risk of ARDS in TBI patients.

This study has several limitations. Firstly, this was a single center database study, and a large number of patients were excluded due to a lack of records of included variables. This selection bias could not be avoided. Most of the excluded patients were mild to moderate TBI. Therefore, this study mainly investigated the incidence of ARDS in severe TBI and the predictive models may be more suitable for use in severe TBI. Future studies with larger sample sizes are worthwhile to evaluate the predictive performance of machine learning models in more generalized TBI patients. Secondly, machine learning models were developed and internally validated using the same dataset from a single medical center. These models should be externally validated in other medical centers in future studies. Thirdly, although the SHAP value is a visualized form of machine learning model, it is still difficult for clinicians to evaluate the risk of ARDS in clinical practice. It is worthwhile to develop a practical application incorporating random forest or adaboost algorithms which could be readily used with an estimated accurate value of ARDS possibility in portable electronic equipment for TBI.

## 5. Conclusions

Machine learning algorithms identified some factors of ARDS in TBI including age, initial systolic blood pressure and heart rate, AIS chest, WBCs, platelets, and INR. The random forest and AdaBoost based models perform efficiently and stably in the prediction of ARDS in TBI patients. These models could help clinicians to evaluate the risk of ARDS in the early stage after TBI, and consequently adjust treatment strategies to prevent the development of ARDS during hospitalizations for TBI.

## Figures and Tables

**Figure 1 medicina-59-00171-f001:**
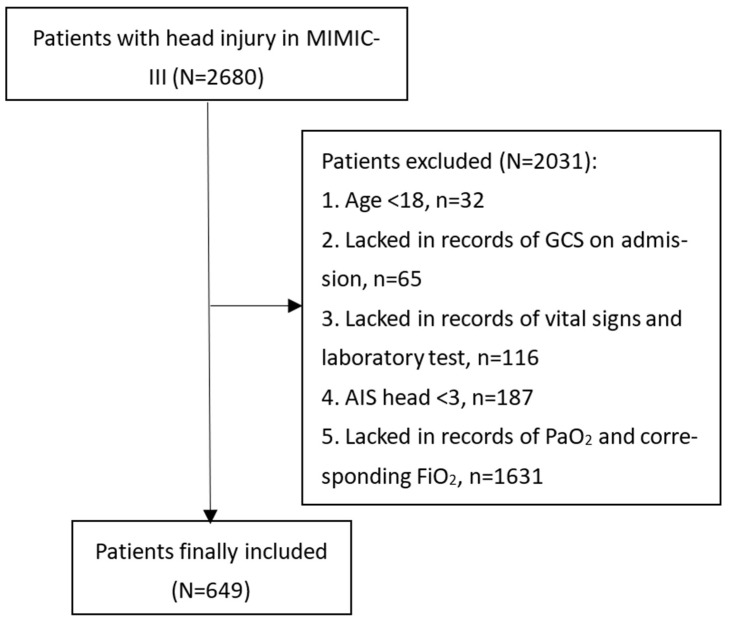
Flowchart of patients’ inclusion. MIMIC, Medical Information Mart for Intensive Care; GCS, Glasgow Coma Score; AIS, Abbreviated Injury Score; PaO_2_, arterial oxygen pressure; FiO_2_, fraction of inspired oxygen.

**Figure 2 medicina-59-00171-f002:**
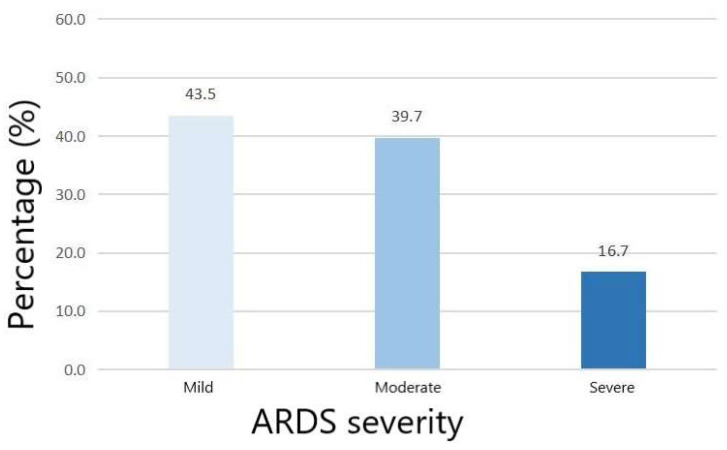
The percentage of ARDS severities in included TBI patients.

**Figure 3 medicina-59-00171-f003:**
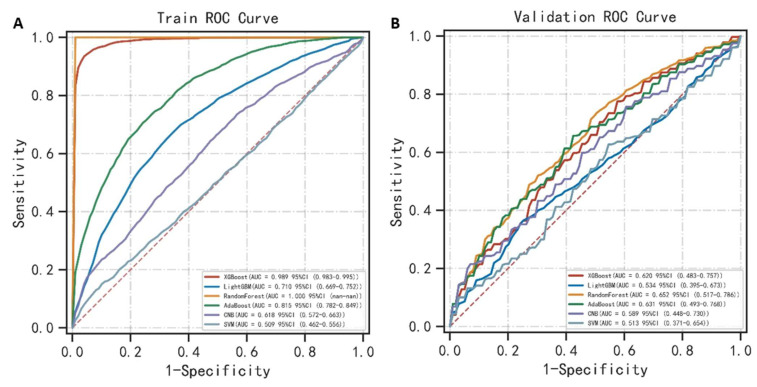
(**A**) Receiver operating characteristics curve of machine learning algorithms for predicting ARDS in the training cohort; (**B**) Receiver operating characteristics curve of machine learning algorithms for predicting ARDS in the validation cohort.

**Figure 4 medicina-59-00171-f004:**
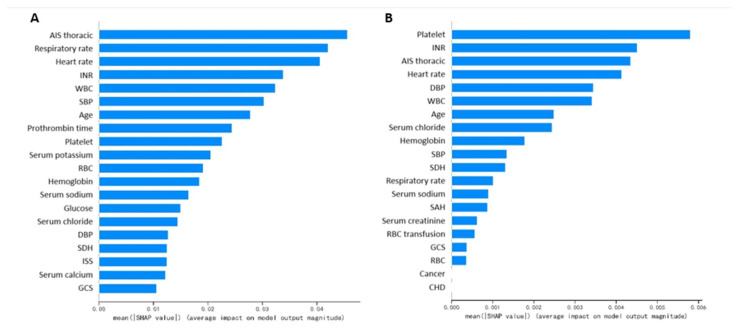
(**A**). Feature importance derived from random forest; (**B**) Feature importance derived from AdaBoost; (**C**) SHAP value of all patients’ outputs in the random forest model; (**D**) SHAP value of all patients’ outputs in the AdaBoost model. AIS, Abbreviated Injury Score; INR, international normalized ratio; WBC, white blood cell; SBP, systolic blood pressure; RBC, red blood cell; DBP diastolic blood pressure; SDH, subdural hematoma; ISS, Injury Severity Score; GCS, Glasgow Coma Scale; SAH, subarachnoid hemorrhage; CHD, coronary heart disease.

**Figure 5 medicina-59-00171-f005:**
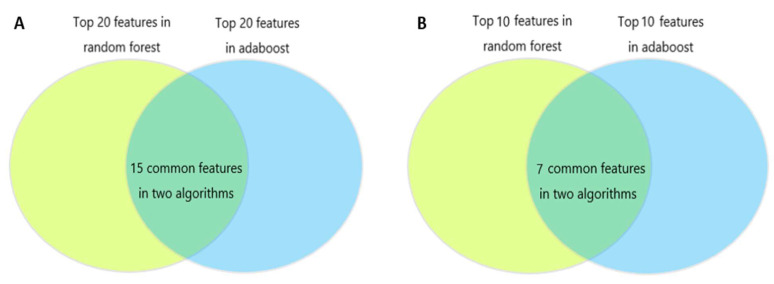
(**A**) Venn diagrams of the top 20 features in random forest and adaboost. There were 15 common features in these two algorithms including platelet, INR, AIS thoracic, heart rate, DBP, WBC, age, serum chloride, hemoglobin, SBP, SDH, respiratory rate, serum sodium, GCS, and RBC. (**B**) Venn diagrams of the top ten features in random forest and adaboost. There were seven common features in these two algorithms including platelet, INR, AIS thoracic, heart rate, WBC, age, and SBP. INR, international normalized ratio; AIS, Abbreviated Injury Score; DBP, diastolic blood pressure; WBC, white blood cell; SBP, systolic blood pressure; SDH, subdural hematoma; GCS, Glasgow Coma Scale; RBC, red blood cell; CHD, coronary heart disease.

**Table 1 medicina-59-00171-t001:** Characteristics of included TBI patients.

Variables	Overall Patients (n = 649)	Non-ARDS Group (n = 328, 50.5%)	ARDS Group (n = 321, 49.5%)	*p*
Age (year)	59.3 (38.8–77.4)	57.2 (31.9–76.1)	62.7 (44.0–78.1)	0.027
Male gender, n (%)	424 (65.3%)	208 (63.4%)	216 (67.3%)	0.300
Comorbidities				
Diabetes, n (%)	93 (14.3%)	47 (14.3%)	46 (14.3%)	1.000
Hypertension, n (%)	187 (28.8%)	85 (25.9%)	102 (31.8%)	0.099
Hyperlipidemia, n (%)	47 (7.2%)	25 (7.6%)	22 (6.9%)	0.706
Coronary heart disease, n (%)	54 (8.3%)	27 (8.2%)	27 (8.4%)	0.934
Liver disease, n (%)	21 (3.2%)	10 (3.0%)	11 (3.4%)	0.786
Chronic renal disease, n (%)	24 (3.7%)	11 (3.4%)	13 (4.1%)	0.638
Malignancy, n (%)	42 (6.5%)	16 (4.9%)	26 (8.1%)	0.095
Vital signs on admission				
Systolic blood pressure (mmHg)	130 (113–146)	131 (114–147)	130 (112–144)	0.584
Diastolic blood pressure (mmHg)	64 ± 17	64 ± 17	64 ± 16	0.811
Heart rate (s^−1^)	84 (71–97)	84 (73–95)	84 (71–98)	0.959
Respiratory rate (s^−1^)	17 (14–20)	17 (14–20)	17 (14–20)	0.299
GCS	6 (3–9)	6 (3–9)	6 (3–9)	0.724
AIS chest	0 (0–3)	0 (0–0)	0 (0–3)	**<0.001**
ISS	20 (16–29)	18 (16–25)	22 (16–29)	**0.009**
Laboratory tests				
WBC (10^9^/L)	13.40 (10.00–18.10)	13.30 (9.90–17.30)	13.50 (10.10–18.80)	0.348
Platelet (10^9^/L)	228 (175–288)	238 (190–292)	221 (166–277)	**0.004**
RBC (10^9^/L)	4.07 (3.60–4.51)	4.07 (3.63–4.48)	4.09 (3.55–4.53)	0.874
Hemoglobin (g/dL)	12.70 (11.20–14.00)	12.80 (11.30–13.90)	12.60 (11.10–14.30)	0.713
Glucose (mg/dL)	149 (121–186)	143 (118–186)	153 (123–185)	0.246
Blood urea nitrogen (mg/dL)	16 (12–22)	16 (12–22)	16 (12–22)	0.275
Serum creatinine (mg/dL)	0.90 (0.70–1.10)	0.90 (0.70–1.10)	0.90 (0.70–1.10)	0.522
Serum sodium (mmol/L)	140 (137–142)	139 (137–142)	140 (138–142)	0.053
Serum potassium (mmol/L)	3.90 (3.60–4.30)	3.90 (3.60–4.20)	3.90 (3.60–4.30)	0.195
Serum chloride (mmol/L)	105 (102–109)	105 (101–109)	105 (102–109)	0.787
Serum calcium (mmol/L)	1.17 (1.06–8.20)	1.19 (1.06–8.20)	1.17 (1.05–8.20)	0.561
Prothrombin time (s)	13.20 (12.60–14.30)	13.00 (12.60–14.00)	13.30 (12.70–14.70)	**0.002**
INR	1.20 (1.10–1.30)	1.10 (1.10–1.30)	1.20 (1.10–1.40)	**<0.001**
PaO_2_ on the first day (mmHg)	228 (141–329)	255 (196–364)	179 (104–289)	**<0.001**
FiO_2_ on the first day (%)	100 (50–100)	100 (50–100)	100 (50–100)	0.874
PaO_2_/FiO_2_ ratio on the first day (mmHg)	304 (190–428)	356 (248–452)	248 (143–361)	**<0.001**
Intracranial injury types				
Epidural hemorrhage, n (%)	174 (26.8%)	103 (31.4%)	71 (22.1%)	**0.008**
Subdural hemorrhage, n (%)	339 (52.2%)	187 (57.0%)	152 (47.4%)	**0.014**
Subarachnoid hemorrhage, n (%)	296 (45.6%)	161 (49.1%)	135 (42.1%)	0.072
Intraparenchymal hemorrhage, n (%)	146 (22.5%)	76 (23.2%)	70 (21.8%)	0.677
Treatments				
RBC during the first 24 h, n (%)	87 (13.4%)	51 (15.5%)	36 (11.2%)	0.105
Platelet during the first 24 h, n (%)	73 (11.2%)	28 (8.5%)	45 (14.0%)	**0.027**
Anticoagulants during the first 24 h, n (%)	156 (24.0%)	79 (24.1%)	77 (24.0%)	0.977
Antiplatelets during the first 24 h, n (%)	5 (0.7%)	2 (0.6%)	3 (0.9%)	0.636
Vasopressor during the first 24 h, n (%)	88 (13.6%)	46 (14.0%)	42 (13.1%)	0.726
Mechanical ventilation, n (%)	591 (91.1%)	299 (91.2%)	292 (91.0%)	0.931
Neurosurgery, n (%)	259 (39.9%)	129 (39.3%)	130 (40.5%)	0.761
Length of ICU stay (days)	5.7 (2.4–12.1)	3.8 (1.9–8.4)	7.3 (3.8–14.7)	**<0.001**
Length of hospital stay (days)	10.1 (4.9–18.5)	7.9 (4.0–15.7)	12.4 (6.2–23.0)	**<0.001**
30-day mortality, n (%)	191 (29.4%)	95 (29.0%)	96 (29.9%)	0.792

GCS, Glasgow Coma Scale; AIS, Abbreviated Injury Score; ISS, Injury Severity Score; WBC, white blood cell; RBC, red blood cell; INR, international normalized ratio; PaO_2_, arterial oxygen pressure; FiO_2_, fraction of inspired oxygen. Bold values indicated *p* < 0.05.

**Table 2 medicina-59-00171-t002:** Performance of machine learning algorithms for predicting the ARDS in the training cohort of TBI patients.

Classification Models	AUC (95% CI)	Accuracy	Sensitivity	Specificity	PPV	NPV	F1 Score
XGBoost	0.989 (0.983–0.995)	0.952	0.947	0.960	0.959	0.946	0.953
Light GBM	0.710 (0.669–0.752)	0.675	0.676	0.682	0.681	0.677	0.674
Random Forest	1.000	0.998	1.000	1.000	1.000	0.997	1.000
AdaBoost	0.815 (0.782–0.849)	0.736	0.724	0.752	0.742	0.736	0.731
CNB	0.618 (0.572–0.663)	0.592	0.694	0.495	0.574	0.624	0.626
SVM	0.509 (0.462–0.556)	0.538	0.253	0.822	0.629	0.534	0.305

XGBoost, extreme gradient boosting; Light GBM, light gradient boosting machine; AdaBoost, adaptive boosting; CNB, complement naïve Bayes; SVM, support vector machine; PPV, positive predictive value; NPV, negative predictive value.

**Table 3 medicina-59-00171-t003:** Performance of machine learning algorithms for predicting ARDS in the validation cohort of TBI patients.

Classification Models	AUC (95% CI)	Accuracy	Sensitivity	Specificity	PPV	NPV	F1 Score
XGBoost	0.620 (0.483–0.757)	0.581	0.654	0.622	0.574	0.589	0.597
Light GBM	0.534 (0.395–0.673)	0.527	0.436	0.727	0.534	0.517	0.448
Random Forest	0.652 (0.517–0.786)	0.542	0.719	0.579	0.767	0.526	0.716
AdaBoost	0.631 (0.493–0.768)	0.599	0.594	0.714	0.606	0.596	0.587
CNB	0.589 (0.448–0.730)	0.567	0.577	0.668	0.555	0.584	0.547
SVM	0.513 (0.371–0.654)	0.524	0.607	0.563	0.619	0.523	0.541

XGBoost, extreme gradient boosting; Light GBM, light gradient boosting machine; AdaBoost, adaptive boosting; CNB, complement naïve Bayes; SVM, support vector machine; PPV, positive predictive value; NPV, negative predictive value.

## Data Availability

The datasets used for the current study are available from the corresponding author on reasonable request.

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
