# Peer review of "Prediction of Acute Respiratory Distress Syndrome in Traumatic Brain Injury Patients Based on Machine Learning Algorithms"

_medicina, 2023, doi:10.3390/medicina59010171_

Round 1
Reviewer 1 Report
This article Is able to review the potential machine learning paradigms to evaluate the program’s capacity to predict the development of ARDS in the case of severe TBI and shows the two superior protocols. The only concerns are that the selection process for the data analysis excludes a large proportion of the cases available within the database. This may reduce the power of the final conclusion.
Hopefully further work with the best fit tools can be done to work with less that perfect case data since this is more reflective of the real life clinical experience.
Author Response
This article Is able to review the potential machine learning paradigms to evaluate the program’s capacity to predict the development of ARDS in the case of severe TBI and shows the two superior protocols. The only concerns are that the selection process for the data analysis excludes a large proportion of the cases available within the database. This may reduce the power of the final conclusion. Hopefully further work with the best fit tools can be done to work with less that perfect case data since this is more reflective of the real life clinical experience.
Response: Thanks for this suggestion. The large proportion of excluded cases is indeed a shortcoming of this study. These patients (1631/2680, 60.86%) were mainly excluded due to lacking records of PaO2 and corresponding FiO2. We supply a table (supplementary Table 1) about comparison between those excluded patients and final included patients. And we found excluded patients were mainly mild to moderate TBI [GCS: 14 (9-15)] and included patients were mainly severe TBI [GCS: 6 (3-9)]. Therefore, the predictive models may be more suitable for use in severe TBI. Additionally, a recent meta-analysis concluded that most of ARDS (79%) developed in severe TBI. This fact emphasized the importance of identifying the risk of ARDS in TBI patients especially those severe TBI. Based on this point, our study achieved this aim though excluding a large part of mild to moderate TBI patients. Whatever, future studies with larger sample sizes are worthwhile to evaluate the predictive performance of machine learning models for ARDS in more generalized TBI patients including mild to moderate TBI. This prospect has been stated in the limitation part of our revised manuscript.
Reviewer 2 Report
REVIEW
|
Manuscript ref no. |
|
medicina-2071988 |
|
Manuscript Title |
|
Prediction of acute respiratory distress syndrome in traumatic brain injury patients based on machine learning algorithms. |
Comment Section:
|
General comment |
|
The objective of the study was to evaluate the performance of several machine learning algorithms for predicting the ARDS in TBI patients. The subject is interesting and it explores one of the areas of neurointensive care that concerns a major public health problem in many countries around the world, such as TBI. The study provides an attempt to understand the importance of the new informatics tools available in medicine, such as machine learning models, to predict ARDS in patients with TBI. However, there are some important limitations to consider. It would be advisable to make the following modifications classified, in my opinion, in major (“very important or almost mandatory) and minor (“suggested”).
|
|
Introduction |
|
Major comments: Page 2, It is mandatory that modifications be made in the last paragraph of the introduction. Although it is clarified that there are no previous studies on the development of machine learning models to predict ARDS in TBI, it is mandatory that you add your hypothesis regarding the potential results to be found in the work. Also, the focus of the objectives should be changed, using "evaluate the performance..." instead of "develop predictive...", as in the abstract. Minor comments: None.
|
|
Methods |
|
Major comments: Page 3, I consider it very important to add the subsections within Methods: Study Design (present key elements of study design early in the paper), Interventions, Definitions, and Ethical aspects. I believe that it allows whoever reads the manuscript to follow an order in the methodological characteristics of the work that makes it much easier to read and understand. I think this would greatly improve the order in which Methods aspects should be described. The current content of the manuscript should be adapted by adding these sections. Page 3, In Figure 1, all acronyms used (MIMIC, GCS, AIS, PaO2, etc.) should be clarified at the bottom. You should consider that each supplementary material in an article (figures, tables) should be able to be read independently of the main manuscript. In addition, you should fix the figure in the part where the point 5 of exclusion is, since the last part cannot be clearly read.. Page 4, Statistical Analysis: Please clarify with which software you performed the analysis. Minor comments: None.
|
|
Results |
|
Major comments: Page 5, At the beginning of the results section, reference should be made again to Figure 1, reporting the number of individuals in each phase of the study. Page 5, Please indicate the number of participants with missing data for each variable of interest. Page 5, Table 1, Please highlight in bold the rows of results with statistical significance (p<0.05). This allows you to better appreciate the relevant results.Page 7, Table 2, Please clarify all acronymus used (PPV, NPV, etc). Page 8, Table 3, Please remove the title of Table 3, that is written twice. Moreover, clarify all acronymus used (PPV, NPV, etc). Page 8, Figure 3, Please remove the title of Figure 3, that is written twice. Page 9, Figure 4, Please remove the title of Figure 4, that is written twice. Moreover, clarify all acronymus used (INR, WBC, etc). Page 10, Figure 5, Please remove the title of Figure 5, that is written twice. Moreover, clarify all acronymus used (INR, WBC, etc).
Minor comments: None. |
|
Discussion |
|
Major comments: Page 10, The first paragraph of the discussion should summarize the key results with reference to the objectives of the study. Although they are mentioned later, the discussion of this type of work should begin with the key results. Page 11, In the paragraph where you address the limitations, I think you should emphasize the great limitation of this work, consisting of the loss of 75.78% of the patients in the sample. Also, I think they should even do a general comparative analysis in the patients excluded from the variables you have, and add it in the results section. This would make it possible to know if the general characteristics of the included and excluded patients were similar, reducing the great bias that the study has.
Minor comments: None. |
|
Conclusions |
|
Major comments: None.Minor comments: None. |
|
References |
|
Major comments: All references must follow the guide of authors of the journal. Most are from the last 5 years, which is adequate.
Minor comments: None. |
|
Other Comments |
|
Title: I think it is right.
|
Additional comments:
Abstract:
The abstract should be adapted to the major modifications suggested in the manuscript.
Keywords:
To make the keywords appear in MeSH, modifications must be made. The keyword “Prediction” should be changed for “Prognosis” or “Prognosis factors”, because it is the way it appears in the MeSH.
Ethical review:
It needs to be added the approval number of the ethics committee.
Author Response
1. Introduction
Major comments:
Page 2, It is mandatory that modifications be made in the last paragraph of the introduction. Although it is clarified that there are no previous studies on the development of machine learning models to predict ARDS in TBI, it is mandatory that you add your hypothesis regarding the potential results to be found in the work. Also, the focus of the objectives should be changed, using "evaluate the performance..." instead of "develop predictive...", as in the abstract.
Response: Thanks for this suggestion. We have updated the introduction part as you recommended.
Methods
2. Major comments:
Page 3, I consider it very important to add the subsections within Methods: Study Design (present key elements of study design early in the paper), Interventions, Definitions, and Ethical aspects. I believe that it allows whoever reads the manuscript to follow an order in the methodological characteristics of the work that makes it much easier to read and understand. I think this would greatly improve the order in which Methods aspects should be described. The current content of the manuscript should be adapted by adding these sections.
Page 3, In Figure 1, all acronyms used (MIMIC, GCS, AIS, PaO2, etc.) should be clarified at the bottom. You should consider that each supplementary material in an article (figures, tables) should be able to be read independently of the main manuscript. In addition, you should fix the figure in the part where the point 5 of exclusion is, since the last part cannot be clearly read.
Page 4, Statistical Analysis: Please clarify with which software you performed the analysis.
Response: Thanks for this suggestion. All acronyms used has been supplemented in our revised manuscript.
We have added the software we performed the analysis in the last paragraph of methods part of our revised manuscript.
3. Results
Major comments:
Page 5, At the beginning of the results section, reference should be made again to Figure 1, reporting the number of individuals in each phase of the study. Page 5, Please indicate the number of participants with missing data for each variable of interest. Page 5, Table 1, Please highlight in bold the rows of results with statistical significance (p<0.05). This allows you to better appreciate the relevant results.
Page 7, Table 2, Please clarify all acronymus used (PPV, NPV, etc).
Page 8, Table 3, Please remove the title of Table 3, that is written twice. Moreover, clarify all acronymus used (PPV, NPV, etc).
Page 8, Figure 3, Please remove the title of Figure 3, that is written twice.
Page 9, Figure 4, Please remove the title of Figure 4, that is written twice. Moreover, clarify all acronymus used (INR, WBC, etc).
Page 10, Figure 5, Please remove the title of Figure 5, that is written twice. Moreover, clarify all acronymus used (INR, WBC, etc).
Response: Thanks for this suggestion. We have revised results of our manuscript according to your comments. The title of figures and tables were not written twice. The subtitle was written for the training cohort and validation cohort, respectively.
4. Discussion
Major comments:
Page 10, The first paragraph of the discussion should summarize the key results with reference to the objectives of the study. Although they are mentioned later, the discussion of this type of work should begin with the key results. Page 11, In the paragraph where you address the limitations, I think you should emphasize the great limitation of this work, consisting of the loss of 75.78% of the patients in the sample. Also, I think they should even do a general comparative analysis in the patients excluded from the variables you have, and add it in the results section. This would make it possible to know if the general characteristics of the included and excluded patients were similar, reducing the great bias that the study has.
Response: Thanks for this suggestion. The first paragraph of the discussion has summarized the key results of the study. The large proportion of excluded cases is indeed a shortcoming of this study. These patients (1631/2680, 60.86%) were mainly excluded due to lacking records of PaO2 and corresponding FiO2. We supply a table (supplementary Table 1) about comparison between those excluded patients and final included patients. And we found excluded patients were mainly mild to moderate TBI [GCS: 14 (9-15)] and included patients were mainly severe TBI [GCS: 6 (3-9)]. Therefore, the predictive models may be more suitable for use in severe TBI. Additionally, a recent meta-analysis concluded that most of ARDS (79%) developed in severe TBI. This fact emphasized the importance of identifying the risk of ARDS in TBI patients especially those severe TBI. Based on this point, our study achieved this aim though excluding a large part of mild to moderate TBI patients. Whatever, future studies with larger sample sizes are worthwhile to evaluate the predictive performance of machine learning models for ARDS in more generalized TBI patients including mild to moderate TBI. This prospect has been stated in the limitation part of our revised manuscript.
5. References
Major comments:
All references must follow the guide of authors of the journal. Most are from the last 5 years, which is adequate.
Response: Thanks for this suggestion. We have updated the reference style according to the guide of authors of the journal.
6. Additional comments:
Abstract:
The abstract should be adapted to the major modifications suggested in the manuscript.
Response: Thanks for this suggestion. The abstract has been adapted in our revised manuscript.
Keywords:
To make the keywords appear in MeSH, modifications must be made. The keyword “Prediction” should be changed for “Prognosis” or “Prognosis factors”, because it is the way it appears in the MeSH.
Response: Thanks for this suggestion. We have changed the “prediction” to “Prognosis factors” in our revised manuscript..
Ethical review:
It needs to be added the approval number of the ethics committee.
Response: Thanks for this suggestion. We have added the approval number of the ethics committee in the methods part of our revised manuscript.
Round 2
Reviewer 2 Report
I have read the article several times because I find the subject very interesting, since it provides one of the initial kicks regarding machine learning to predict ARDS in TBI. The modifications made to the manuscript correspond to what was previously requested. I encourage you to continue this line of research.